## A near-global eddy-resolving OGCM for climate studies

X. Zhang<sup>1</sup>, P. R. Oke<sup>1</sup>, M. Feng<sup>2</sup>, M. A. Chamberlain<sup>1</sup>, J. A. Church<sup>1</sup>, D. Monselesan<sup>1</sup>, C. Sun<sup>2</sup>, R. J. Matear<sup>1</sup>, A. Schiller<sup>1</sup> and R. Fiedler<sup>1</sup>

<sup>1</sup>CSIRO Oceans and Atmosphere, Hobart, TAS, 7001, Australia <sup>2</sup>CSIRO Oceans and Atmosphere, Floreat, Western Australia, 6014, Australia *Correspondence to*: Xuebin Zhang (<u>Xuebin.Zhang@csiro.au</u>)

15

5

25

20

Abstract. Eddy-resolving global ocean models are highly desired for spatially-improved climate studies, but this is challenging because they require careful configuration and substantial computational resources. Model drift, partially related to insufficient model spin-up, imperfect model physics or bias in surface forcing, can be problematic, leading to contamination of climate change signals. In this study, we adapt a near-global eddy-resolving ocean general circulation model, originally developed for

- 5 short-range ocean forecasting, for climate studies. The Ocean Forecasting Australia Model version 3 (OFAM3) is spun up for 20 years, with repeated year 1979 forcing and adaptive relaxation (Newtonian nudging) of temperature and salinity in the deep ocean to an observation-based climatology. In addition, surface heat fluxes from the JRA-55 atmospheric reanalysis are adjusted during the spin-up experiment to minimise excessive net heat uptake in the ocean. In the historical experiment, spanning 1979-2014, a non-adaptive relaxation is applied by repeating the same relaxation rates derived from the last five
- 10 years of the spin-up experiment, and the surface heat flux adjustment diagnosed during the spinup experiment is also maintained. We demonstrate that the historical experiment driven by the JRA-55 reanalysis does not have significant drifts (e.g., as shown by simulated global ocean heat content), and also provides an eddy-resolving simulation of the global ocean circulation over the period 1979-2014. Decadal changes, such as the strengthening of the subtropical gyre circulation, are also reasonably simulated. A biogeochemical model is coupled with OFAM3 to produce patterns of primary productivity and
- 15 carbon fluxes that are consistent with observations. Experiences gained from our numerical experiments will be helpful to other modelling groups who are interested in running global eddy-resolving OGCMs for climate studies.

#### 1 Introduction

Global climate models provide useful information about large-scale climate change and variability in both the past and the future, as shown in the Fifth Assessment Report of IPCC (2013). With relatively coarse resolution (e.g., 1° in the ocean component) global climate models are primarily designed to study large-scale climate change and variability over decades to centuries under different climate scenarios. As a result, mesoscale features in the ocean are often absent in global climate models, however, they are important for studying climate-related variability of ocean boundary currents, eddies, continental shelf and biogeochemical processes. Mesoscale eddies in the ocean play significant roles in heat transport and exchange (Griffies et al. 2015), the ocean kinetic energy budget (Ferrari and Wunch 2008; Schiller et al. 2008), and the supply of nutrients to the upper ocean (McGillicuddy et al. 1998).

Noting that the first baroclinic Rossby radius of deformation in the ocean is in the range of 20~200 km, with smaller values in high latitudes and larger values in low latitudes (e.g., Chelton et al. 1998), the ocean general circulation models (OGCMs), either as stand-alone ocean models or as ocean components in the coupled climate models, need to have a horizontal resolution of  $\frac{1}{4}^{\circ}$  to permit mesoscale eddies, and a higher resolution ( $1/10^{\circ}$  or finer) to resolve them. There have been growing efforts to

30 develop global eddy-permitting and eddy-resolving OGCMs in the past 1-2 decades (e.g., Masumoto et al. 2004; Maltrud and McClean, 2005; Chassignet et al. 2007; Sasaki et al., 2008; Yu et al. 2012; Oke et al. 2013a; Sérazin et al. 2015), often based

5

on experiences gained from setting-up high-resolution basin-scale experiments, e.g., in the North Atlantic (Smith et al. 2000; McClean 2002).

Despite significant increase of computational power, running a global eddy-resolving model over long periods (e.g., > 50 years), is still a major undertaking. In fact, it's quite common that global eddy-resolving models are integrated over 1~2 decades only (e.g., Maltrud and McClean 2005; Yu et al. 2012), with a few pioneering efforts to run global eddy-resolving models over several decades (Sasaki et al. 2008; Sérazin et al. 2015; Griffies et al. 2015). However, the integration time needed for an OGCM to reach thermodynamic equilibrium is much longer (e.g., McWilliams 1996; Griffies et al. 2014). Model drift is a long-standing problem, not only in ocean-only models, but also in coupled climate models, often caused by insufficient model spin-up (e.g., Sen Gupta et al. 2012, 2013; Griffies et al. 2014; Hobbs et al. 2016). For example, in the suite of

- Coordinated Ocean-ice Reference Experiments Phase II (CORE-II) models, with typical grid resolution of 1° (Griffies et al. 10 2014), drifts can persist in some of models even after 300 years of integration. The changes induced by model drift – a model artefact, can be comparable to or ever larger than the real climate changes signals under investigation. The model drift therefore needs to be appropriately understood and quantified (e.g., Sen Gupta et al. 2012, 2013).
- In this study, we modify a near-global eddy-resolving OGCM the Ocean Forecasting Australian Model, Version 3 (OFAM3, Oke et al. 2013a) for climate change applications. OFAM3 and its previous versions (OFAM 1 and OFAM2) were 15 primarily developed for upper-ocean, short-range operational forecasting (Oke et al. 2005; Schiller et al. 2008; Oke et al. 2013b). We implement several practical strategies (e.g., new ways of implementing temperature and salinity relaxation in the deep ocean, and adjusting surface heat fluxes) to run OFAM3 such that after twenty years of spin up from rest OFAM3 does not show significant drifts. Moreover, we show that the new configuration realistically simulates the ocean variability over the
- period 1979-2014. 20

Including a biogeochemical (BGC) model in OFAM3 enables a simulation of primary productivity (PP), nutrient and carbon cycling in the ocean. Ocean productivity is expected to decrease with climate change as increasing stratification of the upper ocean reduces the supply of nutrients (Saramiento et al. 2004). By contrast, previous ocean downscaling of climate projections, have indicated that productivity can increase locally, due to changes in mesoscale variability (Matear et al. 2013). The

mesoscale resolution of OFAM3 allows us to investigate the interaction of the BGC fields with the ocean physics across the 25 near-global model domain.

The paper is organized as follows. In section 2, OFAM3 (Oke et al. 2013a) is briefly introduced, and details of the innovations in the new configuration are explained. The spin-up experiment, forced by repeating year 1979 forcing for two decades, is discussed in Section 3. The historical experiment, over 1979-2014, is described in Section 4. Finally, a discussion and

30 conclusions are presented in Section 5.

15

#### 2 Model setup

As a baseline for this study, we use the most recent OFAM version – OFAM3, described in detail in Oke et al. (2013). In this section, we provide a short description of the model, and focus on the innovations of the new configuration that are motivated to make the model more suitable for climate studies.

- 5 OFAM3, based on version 4p1d of the Geophysical fluid Dynamics Laboratory Modular Ocean Model (Griffies, 2009), is configured to have 0.1° grid spacing for all longitudes and between 75°S and 75°N (~ 8-11 km x 11 km), comprising 3600x1500 horizontal grid points (Fig. 1). It has 51 vertical layers, with 14 layers between the surface and 100 m depth, 19 layers between 100 and 500 m depth, 6 layers between 500 and 1000 m depth, and 12 layers below 1000 m. A partial cell technique (Adcroft et al. 1997) is employed to better represent bottom topography.
- 10 Horizontal mixing, dependent on model state and grid size, is provided by the biharmonic Smagorinsky viscosity scheme (Griffies and Hallberg 2000), while vertical mixing is provided by K-profile parameterization (KPP, Large et al. 1994), rather than the scheme by Chen et al. (1994) adopted in a previous OFAM3 configuration. The effect of barotropic tidal mixing is parameterized by the tidal frictional turbulence, which is combined with the vertical mixing from the KPP (Lee et al. 2006).

The model is forced by 3-hourly Japanese 55-year Reanalysis (JRA-55; Kobayashi et al. 2015). Previous versions of OFAM used surface fluxes of heat and momentum, derived directly from an atmospheric reanalysis (e.g., Oke et al. 2013a). The new

configuration uses bulk formula (Large and Yeager 2004) for wind stress, turbulent sensible and latent fluxes, and evaporation. Total precipitation is based on the JRA-55 reanalysis, while monthly climatologies of river run-offs are from Dai and Trenberth (2002) and Dai et al. (2009). The net global freshwater flux (i.e., sum of evaporation, precipitation and river run-

offs) is balanced at each model time step to ensure consistency with the Boussinesq approximation (Gill 1982). Consequently, the model global ocean volume is conserved and the global mean sea level is always zero (e.g., Griffies et al. 2000).

Model drift in ocean properties in forced experiments is common (e.g., Griffies et al. 2014), as a result of missing interactions with the atmosphere and sea ice, biases in surface forcing, imperfection of model parameterization (e.g., mixing scheme), inaccurate initial conditions, and long-term adjustment processes such as those associated with the thermohaline circulation. As a result, relaxation of temperature and salinity to a climatology, particularly in the deep ocean, is often used to reduce drifts

25 in deep-water properties and to keep the model fields closer to reality. For example, in previous OFAM3 experiments (Oke et al. 2013a), temperature and salinity in the deep ocean (below 2000 m) are weakly relaxed to climatology from the CSIRO Atlas for Regional Seas (CARS 2009 release; Ridgway and Dunn 2003) with a time-scale of one year.

Relaxation is typically added as an extra forcing term to the temperature or salinity equation in the form of Newtonian nudging to prevent the model fields from diverging too much from the observed climatology. Such relaxation is adaptive, and

30 thus depends on the difference between the model state and climatology, with stronger relaxation when differences are larger. Adaptive relaxation represents a time-dependent source or sink of the heat and salt, often obscuring the "true" climate change signals that are of primary interest. However, due to model shortcomings, relaxation is often a common technique used in global eddy resolving OGCMs, including previous OFAM experiments (e.g., Oke et al. 2013a). Here we use a new design by

applying an appropriate non-adaptive relaxation that will minimise model drift, while also allowing the model to simulate deep variability. To achieve this, we apply an adaptive relaxation for temperature and salinity during the 20-year spin-up experiment with repeated year 1979 forcing. We then diagnose a seasonal climatology of it and then repeatedly apply the seasonal relaxation in a non-adaptive manner (i.e., relaxation doesn't depend on model state anymore) throughout the historical experiment. We find that this combination of adaptive relaxation during the spin-up experiment, and non-adaptive relaxation during the historical experiment keeps the deep-ocean close to the observed climatology but allows the climate changes signals

to penetrate to the deep ocean.

In addition to the deep ocean relaxation, temperature and salinity are also relaxed to the CARS climatology from the top to the bottom within a  $5^{\circ}$  buffer zone from the northern boundary in the Northern Atlantic to reduce the impacts from a missing

- Arctic ocean. Within the boundary buffer zone, the relaxation timescale linearly increases from 30 days at the boundary to 365 days in the interior. Sea surface salinity is also weakly restored to monthly CARS climatology with a time-scale of 180 days. In reality, the ocean areas that are under ice don't "feel" the atmosphere. However, because OFAM3 does not include a sea-ice model, the naive application of bulk surface fluxes can lead to problems, e.g., significant heat loss at the surface that triggers massive convection. A solution to this involves the use of the JRA-55 sea ice coverage field to mask the applied atmospheric
- forcing fields. The sea ice coverage field, ranging from 0 (ice-free) to 1 (full-ice), is typically either near 0 or near 1, without extensive intermediate values. Specifically, when the ice cover fraction (a) is greater than 0.1, the air temperature is set to  $1.8 \text{ }^{\circ}\text{C}$  – the freezing point of sea water. Additionally, wind speed, downward shortwave radiation, and downward longwave radiation are scaled by the ice-free fraction (1-a) to give a transition from unmodified atmospheric fields in ice-free conditions, to zero under complete ice cover. Blackbody radiation at the freezing point is also scaled by ice cover (a\*307.4 W m<sup>-2</sup>). Finally,
- the ocean temperature in the model occasionally falls below the freezing point of sea water (-1.8 °C). If this occurs, the temperature is reset to -1.8 °C, a quite common practice for ocean models without an explicit sea ice component (e.g., Yu et al. 2012).

In addition to the physics, OFAM3 includes the World Ocean Model of Biology and Trophic dynamics (WOMBAT; Oke et al. 2013a). WOMBAT is based on a simple nutrient, phytoplankton, zooplankton and detritus model, with the addition of

- an oxygen and carbon cycle. We include WOMBAT in our climate simulations to enable the investigation into how climate changes impact PP and nutrient and carbon cycling. Oke et al. (2013a) includes a detailed description of WOMBAT. In the experiment presented here, the BGC parameters used with WOMBAT are based on Oschlies and Schartau (2005), with extra parameters for the carbon cycle (Law et al. 2015). For the carbon, we include two separate dissolved inorganic tracers: 1) forced with rising atmospheric CO<sub>2</sub> levels (anthropogenic carbon tracer) and 2) forced with atmospheric CO<sub>2</sub> of 280ppm
- (natural carbon tracer). These parameters differ from the implementation of Oke et al. (2013a), and are intended to improve the BGC behaviour in oligotrophic waters. The model BGC fields are initialised with observations. Specifically, the World Ocean Atlas is used to initialise nutrients (phosphorus) and oxygen (Garcia et al 2006a; Garcia et al. 2006b). The Global Ocean Data Analysis Project (GLODAP) product is used to initialise alkalinity and dissolved carbon (Sabine et al. 2004; Key et al. 2004). Phytoplankton is initialised with SeaWIFS observation (NASA, 2014) and zooplankton is initialised as a fraction

This duration spans the era of relevant satellite sea-colour observations.

of phytoplankton (0.05). We also include a restoring term to the deep-water BGC fields below 2000 m, which is employed to ensure the deep-water BGC fields remain realistic. These deep fields (nutrients, oxygen, alkalinity, and carbon) are adaptively relaxed to observations with the same time scale (i.e., one year) and over the same spatial extent as the temperature and salinity fields (see above) in the spin-up experiment. Inclusion of the BGC component of the system is computationally expensive. The BGC fields are therefore spun up for only 15-years – followed by integration in the historical run between 1992 and 2014.

5

#### 3 Spin-up experiment

The correct spin-up of OGCMs is critical for ocean climate applications, and it is important to provide a realistic initial condition (IC) for historical experiments. Only over the most recent decade, the global upper ocean (above 2000 m) has been regularly observed, mainly because of the global Argo float program (Roemmich and Owens 2000), and as a result it became feasible to initialize a model based on in-situ observations. Before the Argo program, it is not a trivial task to get a dynamically consistent IC for hindcast experiments (e.g., McWilliams 1996).

Considering the needs to spin-up the model and to ensure a reasonable IC for the subsequent climate change simulations, we start the model from rest with the initial temperature and salinity fields from CARS climatology (2009 release; Ridgway

- 15 and Dunn 2003), then apply a repeating normal-year type forcing over a long enough period for the model ocean (at least the upper to middle depths) to stabilize. The year 1979 corresponds to the start year of the historical run, and is also chosen as the 'normal year' for the spin-up experiment, because both Pacific Decadal Oscillation (PDO, Mantua and Hare, 2002) and El Niño Southern Oscillation (ENSO, McPhaden et al., 2006) phases are close to near-neutral. The idea of applying normal-year forcing was previously used in the CORE-Phase I (CORE-I) experiment (Griffies et al. 2009).
- Three-hourly forcing from JRA-55 reanalysis for 1979 is applied repeatedly to spin the ocean up. In the spin-up simulation (referred to as "OFAM3-JRA55-spinup" experiment), the ocean rapidly adjusts in the first couple of years, as indicated by the evolution of the global mean kinetic energy (Fig. 2) and global mean ocean heat content (OHC) (Fig. 3). The upper ocean takes only several months for its kinetic energy to reach around ~200 cm<sup>2</sup> s<sup>-2</sup>, and a clear seasonal cycle can be seen from Year 2 of the spin-up experiment (Fig. 2b). In contrast, it takes about two years for the deep layer to reach ~8 cm<sup>2</sup> s<sup>-2</sup> and stay at a
- 25 similar level afterwards (Fig. 2c). The evolution of volume-averaged kinetic energy over the full depth is similar to that of bottom layer, with slightly higher values around 20 cm<sup>2</sup> s<sup>-2</sup> (Fig. 2a).

In the first two years of the simulation the ocean gains heat at a rate of 17.8 W m<sup>-2</sup> and 7.73 W m<sup>-2</sup> in Year 1 and Year 2, respectively (Table 1). If we continue without any heat flux correction, the model ocean state in most regions would develop a warm bias relative to the CARS climatology. To avoid that, in Year 3 the annual mean net surface heat flux from Year 2

30 (7.73 W m<sup>-2</sup>) is removed from the downward longwave radiation. With this heat flux correction, the annual mean net flux in Year 3 is 3.12 W m<sup>-2</sup>, smaller but non-zero. In Year 4, the sum of the mean net heat flux from both Year 2 (7.73 W m<sup>-2</sup>) and Year 3 (3.12 W m<sup>-2</sup>) is removed from the downward longwave radiation, which leads to a net heat flux of 2.48 W m<sup>-2</sup>. Repeating

this approach each year until Year 7, the net heat flux converges to a smaller value of about 1 W m<sup>-2</sup>. From Year 8 to Year 20, the same heat flux correction (-16.925 W m<sup>-2</sup>) is applied, which resulted in gradually reduced net surface heat flux to below 0.5 W m<sup>-2</sup> from Year 15 to Year 20. This value is close to the net surface heat flux (0.4-0.6 W m<sup>-2</sup>) inferred from Argo measurements over the 2006-2014 period (Roemmich et al. 2015).

- 5 The -16.925 W m<sup>-2</sup> heat flux correction is quite large, but is comparable to the large uncertainty and discrepancy of net heat flux from the different atmospheric reanalysis products. For example, the Objectively Analyzed air-sea Fluxes (OAFlux, Yu and Weller 2007) product which objectively blended several popular atmospheric reanalysis products has a mean net heat flux of 30 W m<sup>-2</sup> over 1983-2009, which is much larger than the <1.0 W m<sup>-2</sup> based on observed ocean heat content change in the past several decades (e.g., Rhein et al. 2013). Moreover, air-sea feedback mechanisms through the bulk formulas affect the net 10 heat flux. For example, less heat input from downward longwave radiation initially leads to cooler sea surface temperature
- (SST), which in turn leads to less heat loss from the ocean through turbulent heat fluxes and upward long-wave radiation. So this -16.925 W m<sup>-2</sup> correction does not necessarily imply a net heat flux correction of the same magnitude.

Global OHCs at different depths experience initial shocks in the first several years (Fig. 3). The OHC in the upper 100 m layer stabilizes and repeats its seasonal cycle staring from Year 3, though other upper layers between 100 m and 500 m takes longer time to adjust and stabilize (Fig. 3b). The deep layer below 2000 m experiences an initial cooling in the first couple of

15 longer time to adjust and stabilize (Fig. 3b). The deep layer below 2000 m experiences an initial cooling in the first couple of years, and then stays virtually unchanged, which implies that the adaptive relaxation is efficient at keeping the deep ocean stabilizing (Fig. 3c). There are some small slow adjustments of OHC in 500-1000 m and 1000-2000 m layers. The rate of change of Global OHC, equivalent of net surface heat flux, tends to have a well-repeated seasonal cycle after Year 10 (Fig. 3d). The mean net heat gain during the last five years of spin-up experiment is equivalent to about 0.3 W m<sup>-2</sup> net surface heat flux.

With an average 17.8 W m<sup>-2</sup> heat uptake during the first year, the global mean temperature increases slightly more than 1 °C in the upper 100 m (Figs. 3b, 4a). The changes of global mean temperature over Year 2 and 3 are much reduced, with peak value of ~0.2 °C around 150 m over Year 3 (Fig. 4a). Below 500 m, changes have a typical magnitudes of 0.02 °C over Year 1 (Fig. 4b) and are subsequently about an order of magnitude smaller. Global average salinity also shows larger year-to-year

25

changes in the first few years (Fig. 4c). Unlike surface heat flux, the freshwater flux is not corrected, but a weak surface salinity relaxation (time scale of 180 days) helps to keep upper ocean salinity from deviating too much from its observed climatology. Salinity shows little adjustments (<0.001 psu) below 500 m after Year 3 (Fig. 4d).

The other purpose of the spin-up experiment is to derive a monthly climatology of relaxation of temperature and salinity in the deep ocean and northern boundary, which will be then applied to the historical experiment. Using temperature as an

30 example, the integrated temperature or OHC changes (equivalent surface heat flux in W m<sup>-2</sup>) is the sum of the surface heat flux and relaxation to CARS climatology (Fig. 5). Note that except in the beginning several years the seasonal cycle of surface heat flux calculated by the bulk formulas agrees well with that derived from the OAFlux – an objectively blended dataset of several atmospheric reanalysis products (Yu and Weller 2007) (see solid black curve and dashed green curve in Fig. 4a). Globally, the deep and northern boundary relaxation term (equivalent surface heat flux) has a cooling effect of about -4 W m<sup>-2</sup>

 $^{2}$  during Year 1, and then gradually reduces and oscillates around zero (Fig. 5b). From Year 6 onwards, it repeats itself year after year, with the seasonal cycle mainly originating from the northern boundary relaxation. The annual mean of all temperature relaxation including both the deep ocean below 2000 m and the buffer zone near the northern boundary has a small cooling residual (about -0.1 W m<sup>-2</sup>, cyan curve in Fig. 5b).

- Adjustment timescales in BGC tracers are long and we do not have sufficient computational resources for either an extended BGC spin up or multiple experiments to tune the BGC parameters to the circulation of the OFAM3 experiment. Hence, the objective in this spin-up is to integrate BGC fields past the initial shock and run long enough to reduce the long-term changes in the BGC fields to be less than interannual variability in the historical experiment, and ultimately less than the response to climate change in the future climate projection experiment. The BGC spin-up was done in two parts, a spin-up run of 10 years
- with repeated year 1979 forcing experiment (i.e., Year 11 to Year 20 in the OFAM3-JRA55-spinup run) followed by another 5 years with historical forcing (1979-1984). Fig. 6 shows the time series of simulated PP and natural carbon fluxes from the spin-up. Global PP undergoes an initial rapid change to reach stable value by Year 3 of ~42 PgC yr<sup>-1</sup> (Fig. 5). Separated into different latitude bands (see Fig. 6 for definition), primary productivity displays a similar rapid change that is mostly stable by the end of the simulation. There is a small decreasing trend in the global PP (~0.2 PgC yr<sup>-1</sup>) in the spin-up simulation arising
- from the Northern and Southern sub-tropics (Fig. 6). The global net air-sea natural carbon fluxes undergo a rapid transition over the spin-up simulation as the ocean outgasses. The adjustment time of the global net air-sea flux is longer than that associated with the PP. Separated into latitudes, all latitudes except the north oceans show outgassing that reduces in amplitude over the spin-up. Most latitude bands are stabilising at the end of the experiment. However, the trend in the total carbon flux is affected by interannual variability in the second phase of the spin-up that arises from the tropics and makes it difficult to
- determine whether the carbon fluxes have stabilized.

#### 4 Historical experiment over 1979-2014

The historical experiment is initialized from the final state of the spin-up experiment, and is forced by bulk fluxes using atmospheric fields from the JRA-55 reanalysis for the period 1979-2014 (referred to as "OFAM3-JRA55" run). To keep the impacts of climatological relaxation of temperature and salinity, we repeat monthly values of temperature and salinity relaxation derived from the last five years of the spin-up experiment, in both the deep ocean below 2000 m and the buffer zone near the northern boundary in the North Atlantic. The constant heat flux correction of -16.925 W m<sup>-2</sup>, derived from the spin-

up experiments, is also applied (Section 3).

BGC tracers are integrated over 1992-2014 and are initialised from the final state of the spin-up experiment (refer to Section 3 for BGC spin-up information). BGC tracers were only included for part of the historical experiment because of limited computational resources and to align with the period of high-quality remotely sensed ocean colour observations which began

in 1997.

In this section, the performance of this historical experiment over the past 36 years is examined from the following aspects: mean states, seasonal and interannual variations, meso-scale variability and trends.

#### 4.1 Mean States

#### 4.1.1 SST

- The modelled long-term-mean (1979-2014) SST patterns from the OFAM3-JRA55 experiment closely match the observed patterns from the NOAA 1/4° Optimum Interpolation Sea Surface Temperature (OISST) product (Reynolds et al., 2007) (Fig. 7). These two fields are highly correlated with a spatial correlation coefficient of almost one (0.99). The root mean square (RMS) difference between these two fields is 0.55°C. The model has a mean warm bias of 0.64 °C, with warm biases being located near several major boundary currents (e.g., Gulf Stream, Kuroshio, Benguela Current, Peru Current, Canary Current),
- in the Labrador Sea, and to a lesser degree, along the equator and in the Antarctic Circumpolar Current (ACC). Regions of cool SST bias include the Brazil-Malvinas confluence region, and to a lesser extent, very high northern latitudes and some isolated locations. The cold bias in the Brazil-Malvinas confluence region suggests that the model may be too efficient at advecting cold ACC waters northward after the ACC flows through the Drake Passage. It may also be related to that models tend to have difficulty to simulate the Zapiola Anticyclone (e.g., de Miranda et al. 1999)
- The model-data SST difference fields in all of the major western boundary current (WBC) regions show a narrow band of warm SST bias (Fig. 7c). This bias extends into the Gulf Stream Extension, the Kuroshio Extension, and the Tasman Front, and also into the ACC associated with the Agulhas Extension, in the South Indian Ocean. The narrow warm SST bias in each WBC runs parallel to the coastline until the point where each WBC separates from the coast. This systematic difference could indicates that either the model WBCs transport too much heat poleward (or that the model WBCs transport more heat near the
- surface) or that the sharp SST gradients in the WBCs are better represented in our model simulation than the coarse-resolution OISST observations (compare Fig. 7a, b).

As noted above, several regions known for producing strong and persistent upwelling, including the Benguela Current, Peru Current, the Canary Current, and the equatorial eastern Pacific, are characterised by a warm bias. This systematic difference may be explained by the relatively coarse (1.25°) resolution of the surface wind forcing from JRA-55. At this resolution, the

25 spatial variability of coastal winds is not fully resolved and may explain the modelled bias in regions of strong upwelling.

#### 4.1.2 Sea level

The model represents long-term mean sea level very well (Fig. 8), matching the  $1/2^{\circ}$  observational dataset by Maximenko et al. (2009), which is based on satellite altimeter, in-situ measurements and a geoid model. The RMS difference between model and observations is 0.089 m and the spatial correlation is 0.99 globally. Their global means are almost the same, with model

being -0.002 m lower than observations (averaged over the overlapping domain). Mean sea level from model and observations reflect the upper ocean circulation (Fig. 8). There is a strong negative zonal sea level gradient along the equator in the Pacific,