# Peer review of "A near-global eddy-resolving OGCM for climate studies"

_Geoscientific Model Development, 2016_

## Referee Comment (RC1) · Anonymous Referee #1 · 3 Mar 2016

Summary of key results

The latest version of the Ocean Forecasting Australia Model (OFAM3), a near-global high-resolution ($0.1°$) ocean model is described, based on GFDL MOM v4p1d. Its grid extends from $75°S$ to $75°N$. There is no sea ice, but the model includes the WOMBAT biogeochemical model. The model is forced by the JRA55 climatological surface fields. It is first run for twenty years with a repeat of the 1979 surface climatology to generate a three-dimensional restoration flux, based on that needed to maintain the model fields close to climatology in the first two decades. This flux is then applied to the ocean model to reduce drift during the subsequent integration with interannual forcing up to 2015, but allows the climate signal implied in the forcing to be reproduced.

General style

The paper is overall quite well written and structured, but it is occasionally evident that the text is for the most part written by someone whose first language is not English, so the manuscript would be benefit from further proofreading to check grammar and usage.

Specific corrections

P2, L30: Any list of recent eddy-resolving ocean configurations should include a reference to the latest UK 1/12° NEMO: Marzocchi et al, 2015: Journal of Marine Systems, 142. P4, L19: This is a common, but incorrect, use of the term "Boussinesq approximation". The latter, as (for example!) defined in Gill, 1982, refers to the neglect of density differences except where these imply pressure difference. This is not equivalent to ensuring constant volume.

General comments

The reasons for some of the choices made in creating the OFAM model configuration are puzzling to me, and need to be explained clearly in the paper. The development of a global ocean-only model whose grid reaches 75°S (most of the Antarctic coastline) and has biology but no ice model, and which excludes most of the Arctic, leads to potentially severe limitations on the general applicability of the model to climate studies, and needs to be clearly justified. Such a model may well be useful for understanding recent climatic variability in the midlatitudes and tropics, providing that enough confidence exists in the forcing dataset used (which is not the case before the satellite era), but cannot inform about the interactions between the high and mid latitudes, and certainly can say nothing about the climatically highly important polar regions. For this reason I believe it will have limited interest for the climate science community.

Would it not have been more logically consistent to develop the model from the start as a coupled system, including a fully global ocean? A non-global ocean model cannot directly be reframed as a free-running coupled climate model. The Discussion and Summary Section mentions the intention of re-running the model with the merged out-

put of an ensemble of CMIP5 coupled models, instead of the JRA55 forcing dataset. This is, of course, perfectly feasible, but would have the severe disadvantage of omitting the interannual and decadal climate variability, which would be averaged out in the ensemble mean.

The method of applying the fluxes equivalent to the initial drift of the model is evidently effective in this case in reducing drift on decadal timescales, and is an interesting way to address the thorny question of spinup in climate models. This has of course possible drawbacks, however: specifically, an implicit assumption that the drift is due to model deficiencies, rather than to errors in the applied surface fields or in the way the lateral boundary conditions are applied. In particular, the need for strong restoration of temperature and salinity at the northern boundary has serious implications for the performance of an eddy resolving ocean model, since there is an abrupt mismatch between the resolutions of the model grid and of the forcing data: the consequences for the representation of boundary currents, which are vital for the exchange between the Arctic and the rest of the oceans, are not clear, and should be discussed.

I have difficulty with the recommendation at the end of the paper, namely that the climate modelling community "can consider adopting the approach described in this study as an efficient short-term solution, at the same time also develop more sophisticated methods to address this important problem of model drift". This method was indeed often used in the coupled climate models used at the end of the last century, and was called "flux correction", though this was mainly (but not exclusively) restricted to surface fluxes. By around 2000, though, climate models had improved to a state where they had realistic enough surface fields that they were able to be integrated without flux correction (for example HadCM3, Gordon et al, 2000). Admittedly, interior drifts remain an issue in coupled models, but work is ongoing to reduce this; for instance the use of isopycnal-type ocean models (e.g. Megann et al, J.Clim 2010, Dunne et al J.Clim 2011), that reduce spurious numerical mixing. Do the authors consider that interior drift is a serious enough problem to merit such an invasive engineering fix? This is not discussed in the paper. In my opinion, the proposed use of flux correction in standalone climate models would be a serious step backwards. In any case, the question needs to be posed of the robustness of the tuning of the correction fluxes: would they still be appropriate for models used for future climate projections?

For publication in GMD, the intended application of the model described should be clear, and I am not convinced that is the case here: the introduction section of the paper does not make the case strongly enough for the utility of the OFAM model in climate studies. In particular, I do not really understand the inclusion of the word "forecasting" in the name of the model, since it is incapable of advancing beyond the limit of current forcing datasets. I would certainly like to see a clearer exposition of what the model is actually useful for.

Recommendation

In summary, although the paper is overall well structured and well written, the lack of clear justification for the design of the model, nor of any clear, plausible statement of its intended application, means I cannot recommend publication in GMD of the manuscript in its current form.

---

## Referee Comment (RC2) · S. M. Griffies (Referee) · 6 Mar 2016

This manuscript provides a summary of features found in a mesoscale eddy rich ocean simulation forced by JRA55 atmospheric reanalysis. The manuscript is well written and offers useful diagnostics for others to compare/contrast. It is suitable for GMD, and its publication should ultimately occur. However, there are some overall minor changes needed to bring the manuscript into a more suitable format. So long as the authors address all reviewer comments, and I trust they can, then I recommend this work be published in GMD.

General comments:

\*\*Remove "eddy resolving" everywhere

I strongly rebel against the term "eddy resolving". That term is not justified here, nor even defined. The ocean mesoscale, as defined by the 1st baroclinic Rossby radius, has a non-homogeneous eddy length scale spanning from 1km on the shelves of the high latitudes, to 100km in the low latitudes (see Figure 1 in Hallberg, Ocean Modelling 72 (2013) 92–103). So even from the 1st Rossby radius perspective, this model is not "eddy resolving" everywhere. What about higher baroclinic modes?? And what about submesoscale eddies, and then nonlinear gravity wave "eddies", each of whose features reach down into the sub-kilometre range?

Furthermore, there is no study showing numerical convergence in the mesoscale resulting from a model that has a resolution equivalent to the first baroclinic Rossby wave. What in fact do we need to resolve in order to claim we are "resolved"? Is it just a linear baroclinic wave itself? Or the flux convergences? What fluxes? PV, heat, salt, momentum, etc? The question of what defines "eddy resolving" is not closed, so please avoid this sort of terminology. You have a model that richly represents nonlinear mesoscale features, and you are exploring elements of the simulation. But you cannot claim to be "eddy resolving" by any stretch. Period.

So please drop the pretentious and ill-defined "eddy resolving" term \*\*\*everywhere\*\*\* in your manuscript. Instead, be more explicit and honest by using language such as "mesoscale eddy rich".

\*\*Why no sea ice model?? It needs to be better motivated, even if it is the result of a "model of opportunity" exercise.

\*\*change "and also" to "and" throughout the paper. page 2, line 12 page 7, line 17 page 12, line 22

Specific comments:

page 3, line 12: "ever" should be "even"

page 3, line 23: "Saramiento" should be "Sarmiento"

[Figure]

page 4, line 9: The MOM code uses partial bottom cells based on the work of Adcroft et al (1997) as well as the following paper, which should also be cited:

@Article{PacGnan1998, author = "Ronald C. Pacanowski and A. Gnanadesikan", title = "Transient response in a $z$-level ocean model that resolves topography with partial-cells", journal = "Monthly Weather Review", year = "1998", volume = "126", pages = "3248-3270" }

page 4, lines 18-20: Boussinesq models need not retain a constant volume. They are quite able to increase or decrease sea level through water addition or removal. For example, see the paper

@Article{Lorbacher_etal2012, author = "K. Lorbacher and S. J. Marsland and J. A. Church and S. M. Grif\/f\/ies and D. Stammer", title = "Rapid barotropic sea-level rise from ice-sheet melting scenarios", journal = JGR, year = "2012", volume = "117, C06003", doi = "doi:10.1029/2011JC007733", }

The reason that groups often keep the net water flux equal to zero over the globe is to reduce model drift. It has NOTHING to do with the Boussinesq approximation.

page 6, line 8: What is "The correct spin-up of OGCMs"? Do you presume to have example of such? I for one have been on a 25 year quest for this method... :-)

page 7, line 13: I prefer to think of the global OHC as the global volume mean heat content. However, as used here, the authors refer to the horizontally integrated heat content as a function of depth. I suggest a more suitable language is useful to avoid reader confusion.

page 7, line 14: "staring" should be "starting"

page 7, line 18: there seems to be a missing number on this line. It presently reads "change of Global OHC...". The blank must be a number, but that number is missing.

Figures: I encourage placing more statistical information on each of the figures or their

captions, so to better allow them to be self-contained. Many statistics are noted in the text, and they should also be placed along with the figure. Additionally, for the maps, it would serve the reader well to also provide a zonal mean of the biases to better identify the latitude where the biases are localized. All of this information is useful for others aiming to perform quantitative comparisons to your work. Merely showing the maps is insufficient.

page 9, line 19: "indicates" should be "indicate"

page 9, line 24: JRA-55 is available at a resolution of 55km as well as 1.25 degrees.

http://rda.ucar.edu/datasets/ds628.0/

You can examine this hypothesis concerning upwelling by moving to the finer resolution data.

page 10, section 4.1.4: Please show the maths for how you computed the Sverdrup/Island rule transport shown in Figure 10. Others may wish to repeat this calculation, and your method should be clearly documented. Reliance on literature is not needed, since you can summarize the method in a few lines of words and maths.

page 11, line 2: The meridional overturning streamfunction is not "zonally averaged", in which case the units would be m2/sec. Instead, it is "zonally integrated". Please correct.

page 14, line 13: More discussion is needed regarding this point. It contrasts with points that others have made. Namely, even though OISST is 1/4 degree, it is sampling the real ocean with real ocean variability. This situation is quite distinct from a 1/4 ocean model that only admits variability at the 1/4 and coarser range. For more on this point, please see the paper Levy et al, Ocean Modelling, 48, 2012, Pages 1–9.

page 15, line 9-10: I am unsure why the bulk formula should produce less wind stress than the wind stress directly provided by JRA55. What are the issues? Perhaps it is due to use of (U_atmos - U_ocean) to compute the stress??

**GMDD**

page 15, line 30: "Domingues"

page 16, line 3-4: I have trouble with your statement that "no observational data are assimilated in our historical experiment." And later the statement "nearly free OGCM". Your results are an achievement. But there are huge limitations.

–What about the non-adaptive forcing in the ocean interior that reduces drift (page 16, line 20)?

–What about the adaptive surface nudging?

–What about the truncated northern and southern domain boundaries?

–What about the absence of sea ice?

All of these limitations are helpful to reduce model drift. The high latitudes in particular are very very tough in ocean-ice models. See the Griffies et al, 2009 CORE paper for mechanisms leading to drift, with these mechanisms largely eliminated by truncating your model domain and removing sea ice.

page 16, line 29: "...associated with the..."

page 18, line 11: Suggested edit: "Despite good model performance for certain diagnostics/metrics, we fully acknowledge many caveats and limitations."

page 18, line 17: "combing" should be "combining"

page 27, line 4: "annual"

page 28: this table needs observational estimates for the reader to have any idea of how relevant the model transports are!

page 29, Figure 1: I can barely read the numbers places on the map. They will undoubtedly come out even less visible in print.

page 31, Figure 3 caption: What does "Equivalent surface heat flux" referred to here? Please define.

page 32, Figure 4: I suggest rotating these panels clockwise by 45degrees, so that a full profile for temperature and salinity sit atop one another.

page 33, Figure 5: Again, what is "equivalent surface heat flux"???

page 34, Figure 6 caption: penultimate domain should be "north subtropical", not "south".

page 35, Figure 7: place zonal mean bias next to bottom panel, and make note of other statisticis in the figure caption.

page 36, Figure 8: difference map should have smaller colour range than raw fields, in order to better view the biases. Also add statistics information to the caption.

page 41-42, Figures 13,14: The BGC figures use black land mask, whereas other figures use grey. Please be consistent.

page 50, Figure 22 caption: "Domingue"

END OF REVIEW

---

## Referee Comment (RC3) · Anonymous Referee #3 · 7 Mar 2016

I find little merit in this manuscript and feel it is unsuitable for publication. The paper describes a collection of ad hoc fixes and patches to the forcing of a high-resolution global model to compensate for shortcomings in the formulation of the problem (non-global, lack of sea-ice). The methods described are typical of those used in the 1980's and 1990's when available surface forcing products were far inferior. The apparent goal of the authors is to share "Experiences gained from our numerical experiments . . ." (pg 2 line 15). I cannot agree that this experience is something any other group should be following in the 21st century.

There is no innovation in ocean model development described in this paper. There are some changes made relative to the model formulation described in the previous paper by Oke et al (2013), but none of these are new to the modeling community (e.g. the incorporation of the KPP mixing scheme). Further, the authors do not described the

sensitivity of the solutions to these changes in formulation.

The main argument for "innovation" in the manuscript is in the adjustment of the surface forcing to constrain global energy balance and addition of fictional forcing in the deep ocean to limit drift. The procedures are described as they were implemented without discussion of their merits relative to other alternatives, nor of why the particular choices were made. In the case of the surface flux adjustment, an obvious and simpler method would be to assess the global imbalance in the JRA-55 forcing using observed SST in the bulk formula prior to model integration and subtract the necessary correction a priori. What is the advantage of the iterative approach described in the paper? Why was the adjustment applied to long wave down flux? Why not reduce shortwave down? There is no rationale provided for the choices made. In the case of the deep restoring, the authors imply that this is common practice in high resolution modeling, but only reference their own work as an example. I cannot think of a single study since the pioneering work of Semtner and Chervin in the 1980s that has used deep restoring in a forward integration of a high-resolution model. The main "innovation" the authors claim is the introduction of "non-adaptive" restoring. This is not a new idea. The same basic ideas are described in for example Eden et al JPO, 34, 701-719, 2004 (and likely a number of other papers).

Beyond the generally poorly motivated paper, the manuscript is quite superficial in its assessment of the solutions, and largely speculative when it comes time to attribute bias. Phrases like "may explain" or "could indicate" appear throughout when a more quantitative evaluation at a process level is called for. The aspects of the manuscript dealing with the BGC simulation seem to have been added as a complete afterthought.

DETAILED COMMENTS

pg 4 / lines 14-16. What is the implied global net heat flux for the JRA-55 product when using observed SST? This is a direct indication of the expected drift.

pg 4 / line 19: Why should the hydrologic cycle be balanced instantaneously? Cannot

water be stored on land seasonally?

pg 4 / line 19: Global volume conservation is unrelated to the Boussinesq approximation

pg 4 / line 24: if relaxation is "often used" provide some references of examples other than your own model

pg 4 / line 33 : it is not a common technique (as above - provide some references)

pg 5 / lines 8-22: This is a slew of no-conservative physics. There needs to be a fuller demonstration of its impact not just on global measures, ut on the local structure of the solution. What is the spatial distribution of the restoring term? What spurious heat transports are implied by it? How does it impact the mesoscale?

pg 5 / line 8 : what is the "correct" spin-up. The ocean state in 1979 was not in equilibrium with the forcing in 1979.

pg 6 / line 15-20: How is the year-end discontinuity handled?

pg 6 / lines 27-bottom: There is no justification provided for this ad hoc procedure. (see above)

pg 7 / line 12 "does not necessarily imply a net heat flux correction" Of course it does. That is what you have constructed it to do!

pg 7 / line 32-35: Why does it agree better later in the run? The SST has diverged further from the observed initial conditions. You should be comparing to OAflux for 1979, not its climatology.

pg 9 / line 12 : "model may be too efficient" This is not a meaningful diagnosis and is purely speculative. Could be equally well attributed to any other process.

pg 9 / line 18: "This systematic difference ..." The biases in the Gulf Stream and Kuroshio appear well to the north of separation points, an indication of poor boundary

current separation. This is consistent with the biases int he SSH as well.

pg 9 / line 29: "Their global means are almost the same ..." By construction - you formulated to model to keep global mean sea level constant!

pg 10 / line 28 " "we do not repeat the detailed comparison" Then you don't need the detailed Table.

pg 11 / line 10 : A more useful and critical comparison would be against the vertical structure of the RAPID overturning.

pg 11 / line 27 : They appear more dissimilar than similar to me. The model is completely missing most of the upwelling productivity and vastly overestimates the equatorial productivity.

pg 13 / line 21 : as above - they are more dissimilar than similar. The largest observed variability is in the upwelling areas, not the central Pacific.

pg 14 / line 14: "its possible that ..." There is no need for speculation here. Smooth the model solution to the length scales used in the OI procedure for the observations and compare the smoothed fields.

pg 14 / line 22 : How can the mean upwelling be simulated poorly, but the variability be well reproduced?

pg 15 / line 15: What are we supposed to be seeing in the single-month plots? The eddies will all be in different places for a forward model run.

pg 15 / line 31 - : The full depth OHC is completely determined by the surface heat flux, so it is not surprise these are similar. Why are there no comparisons with the vertical structure from the reanalysis products?

———————————————

---

## Short Comment (SC1) · 10 Apr 2016

Y. Yu

yyq@lasg.iap.ac.cn

The manuscript by X. Zhang et al. presented very comprehensive diagnosis from a high resolution OGCM forced by JRA55 reanalysis, which provided very helpful information for many researchers on modeling, climate change etc. Therefore, it is suitable for GMD journal, and I would like to recommend it to publish. Here are some detailed comments as follows.

1. The mechanism of 1998-2004 Hiatus remains unclear, the numerical experiments in this manuscript provide an important opportunity to understand the Hiatus. For example, how does the model reproduce the basic characteristics of Hiatus such as temperature anomalies in the surface, subsurface and deep ocean, and is there any relationship between hiatus and AMOC? 2. As you mentioned in page 4, bulk formula as

[Figure]

suggested by Large and Yeager( 2004) are applied to calculate the turbulent heat flux and moment flux at the sea surface? However, I am very curious that you have to apply for a large heat correction more 16W/(m*m). Is it resulted from overestimated downward radiation flux in JRA55? 3. Because of large heat flux correction, I suggested that the authors had better show comparison of zonal mean shortwave and longwave radiation from JRA-55, OAFLUX or other available observation. 4. How is the temperature change defined in the Figure 4 ? Is it defined as difference in temperature between the last day and the first day for a given model year? 5. Figure 7, there are significant warm bias at high latitudes in the North Atlantic. Is it due to surface boundary condition, lateral restoring boundary condition or something else? 6. The caption in Figure 9 is "Mean Eddy Kinetic Energy ...", is this correct? 7. Figure 10, I suggested that "mean stream function" in the caption should be replaced with "mean barotropical stream function". 8. The authors calculated simulated MKE and EKE using surface currents, but observed MKE and EKE using sea surface height. If both simulated and observed EKE and MKE are estimated from sea surface height, the comparison may be more reasonable.

---

## Author Comment (AC1) · 7 Jun 2016

**Responses to Reviews of manuscript "A near-global eddy-resolving OGCM for climate studies" (gmd-2016-17) by** X. Zhang, P. R. Oke, M. Feng, M. A. Chamberlain, J. A. Church, D. Monselesan, C. Sun, R. J. Matear, A. Schiller and R. Fiedler.
(The reviewer's comments are in back and our responses are in blue)

We would like to thank all referees for their critical comments, which helped us to clarify and improve the manuscript significantly.

Given the comments, we feel it would be beneficial to first provide two general responses to clarify the motivation and intended application of our model, and the appropriateness of our modelling technique, before we address each reviewer's comments point by point.

**General Responses:**
**#1. What is the motivation of our model experiment? The intended application?**

High-resolution ($1/10^o$ or finer) OGCMs are desired to study ocean's responses to climate change, since they can provide "eddy-resolving" (or "eddy-rich" to be strict) representation of physical processes, which are absent in coarse-resolution ($\sim 1^o$) OGCMs. It's still challenging in ocean modelling to run eddy-resolving or eddy-rich global OGCMs over long periods (> 50 years). Further, it is anticipated that by resolving eddies the future impact of climate change on the ocean environment and marine biogeochemical cycles may differ from coarse resolution models that do not resolve eddies.

In this paper, we adapted an existing $1/10^o$ OGCM (OFAM3) and developed several strategies to carry out long period simulations. We demonstrated that after twenty years of spin-up from rest our historical experiment (1979-2014) can realistically represent ocean variability and change at $1/10^o$ resolution, driven by the JRA-55 Reanalysis surface forcing. In particular, we came up with a new and practical way to deal with model drift, through applying weak climatological restoring of temperature and salinity non-adaptively (i.e., independent of model states) in the deep ocean. In the historical experiment over 1979-2014, the weak restoring effectively helps maintain the deep-ocean close to the observed climatology, without contaminating the climate change signals of interest.

The strategies developed in our study should also in principle be applicable for other similar ocean modelling experiments, either basin set-up or global set-up.

In this study, we demonstrated a feasible way to deal with two long-existing and challenging issues in ocean modelling: 1. how to spin-up an eddy-rich ocean model? 2. how to control or minimize model drifts in order to study climate change signals? Dealing with these two issues are fundamental to any modelling efforts related to ocean climate change studies (such as ocean heat uptake). What we achieved with our modelling experiments should be a welcome contribution to the modelling community. An excellent simulation of ocean heat content change over 1979-2014 without data assimilation is a very nice achievement. In fact, it has already drawn attention from ocean temperature/heat reconstruction community who want to use our modelling result as a test bed to examine various mapping and reconstruction methodologies (Matt Palmer, personal communication, 2015).

However, we are very aware that our model set-up was not perfect, and an approach that address the root problem of model drift would be ideal. Specifically, we stated that our approach could be an efficient choice in the short term (refer to the next response), and suggested that more model development is needed to address above challenging issues.

Here we provide an example of **intended application**. Using the same configuration as described in this manuscript, we have carried out a further climate downscaling experiment ("future experiment") over 2006 to the end of $21^{st}$ Century, driven by merged atmospheric forcing which

combines the high-frequency part from JRA-55 reanalysis product with the long-term climate change part from the CMIP5 ensemble. Such experiment requires that the OGCM can be integrated over about 100 years and doesn't have larger drifts. Our model set-up satisfied this requirement very well, as we found in the recently-finished future experiment (please see our next General Response and refer to Fig. A - ocean heat content plot there). We are currently drafting a separate manuscript based on findings from the future climate experiment.

We have now revised the introduction to incorporate this general response.

**#2. Whether flux correction techniques should be abandoned completely or utilized wisely?**

Referees 1 & 3 had some concern about our usage of flux correction (or restoring). We discussed this point in the Discussion Section of the paper, and cited a recent example of using similar "correction" technique by Vecchi et al. (2014).

*"The design of combining adaptive relaxation of temperature and salinity during the spin-up and non-adaptive relaxation during historical experiment is novel, though its underlying idea has been known to the modelling community, mostly applied to air-sea fluxes. For example, surface flux correction was once a common practice for coupled climate models (e.g., Sausen et al. 1988), e.g., helped to represent ENSO variability better (Roeckner et al. 1996), but it became less common in recent years. Nonetheless, Vecchi et al. (2014) recently rejuvenated such method to correct systematic ocean biases through flux correction and achieved better performance in tropical cyclones forecasting."*

Vecchi et al. of NOAA/GFDL is not the only group who are still applying correction in their model experiment. Magnusson et al. (2013) of ECMWF found that for their seasonal and decadal forecast, flux-correction method gives better forecasting results than the full initialisation and anomaly initialisation methods. In particular, they argued that flux-correction has its own advantage and should not be taken off the table from a pragmatic point of view. Mojib Latif's group at GEOMAR uses various flux-corrections in their experiments with the Kiel Climate Model (e.g., Ding et al. 2015; personal communication with. M. Latif). Ding et al. (2015) found only a flux-corrected model experiment can capture the equatorial Atlantic interannual variability reasonably well, while uncorrected experiment cannot.

Based on the fact that some leading centres are still using this technique, what we have achieved with our model experiment should not be regarded as a "serious step backwards", but rather a practical application of existing idea/methodology.

[Figure]

**Figure A.** Global ocean heat content ($10^{25}$ Joule) of the model simulations for spin-up (black), historical (red), future (green) and control (blue) experiments. Spin-up and historical experiments were described in this manuscript. With the same configuration as the historical experiment, the future experiment is initialized from the ocean state at the end of 2005 of the historical experiment, and is driven by climate change forcing derived from

CMIP5 models. The control experiment, initialized from the final state of the spin-up experiment, is integrated with repeated year 1979 forcing and non-adaptive relaxation.

As mentioned in the Discussion Section, using the same configuration as described in this manuscript, we run a further experiment to downscale ocean climate changes from 2006 to the end of 21$^{st}$ century. We also run a parallel control experiment from 1979 to the end of 21$^{st}$ century, which has repeated 1979 year forcing and non-adaptive relaxation, to quantify any residual drift. Figure A shows the ocean heat content (OHC) from all four experiments (spin-up, historical, future and control). The lack of a discernable trend in the OHC from the control experiment (with repeated forcing, non-adaptive restoring in the deep ocean) demonstrates that our experimental design was successful.

We have now added more information about using "correction" in the introduction, and cited recent relevant publications.

**References:**
Ding, H., Greatbatch, R. J., Latif, M., and Park, W.: The impact of sea surface temperature bias on equatorial Atlantic interannual variability in partially coupled model experiments, Geophys. Res. Lett., 42, 5540–5546, doi:10.1002/2015GL064799, 2015

Magnusson, L., Alonso-Balmaseda, M., Corti, S., Molteni, F., and Stockdale, T.: Evaluation of forecast strategies for seasonal and decadal forecasts in presence of systematic model errors. Clim. Dyn., 41, 2393-2409, 2013.

---

## Author Comment (AC2) · 7 Jun 2016

**Responses to Reviews of manuscript "A near-global eddy-resolving OGCM for climate studies" (gmd-2016-17) by** X. Zhang, P. R. Oke, M. Feng, M. A. Chamberlain, J. A. Church, D. Monselesan, C. Sun, R. J. Matear, A. Schiller and R. Fiedler.
(The reviewer's comments are in back and our responses are in blue)

**Referee Comment by anonymous referee #1:**

**Summary of key results**
The latest version of the Ocean Forecasting Australia Model (OFAM3), a near-global high-resolution (0.1°) ocean model is described, based on GFDL MOM v4p1d. Its grid extends from 75°S to 75°N. There is no sea ice, but the model includes the WOMBAT biogeochemical model. The model is forced by the JRA55 climatological surface fields. It is first run for twenty years with a repeat of the 1979 surface climatology to generate a three-dimensional restoration flux, based on that needed to maintain the model fields close to climatology in the first two decades. This flux is then applied to the ocean model to reduce drift during the subsequent integration with interannual forcing up to 2015, but allows the climate signal implied in the forcing to be reproduced.

**General style**
The paper is overall quite well written and structured, but it is occasionally evident that the text is for the most part written by someone whose first language is not English, so the manuscript would be benefit from further proofreading to check grammar and usage.

Thank you for your positive comments and a good suggestion. We have asked our native-speaking co-authors to double check the grammar and usage in the manuscript and we hope it reads well now.

**Specific corrections**
P2, L30: Any list of recent eddy-resolving ocean configurations should include a reference to the latest UK 1/12_ NEMO: Marzocchi et al, 2015: Journal of Marine Systems, 142.

We cited one NEMO based paper (Sérazin et al. 2015) and added the Marzocchi et al (2015) citation.

P4, L19: This is a common, but incorrect, use of the term "Boussinesq approximation". The latter, as (for example!) defined in Gill, 1982, refers to the neglect of density differences except where these imply pressure difference. This is not equivalent to ensuring constant volume.

Thank you for clarifying our statement, the paragraph has now been modified to
"The model adopts Boussinesq Approximation (Greatbatch 1994). The net global freshwater flux (i.e., sum of evaporation, precipitation and river run-offs) is balanced at each model time step, thus the global mean sea level is kept constant."

Reference:
Greatbatch, R. J., A note on the representation of steric sea level in models that conserve volume rather than mass, J. Geophys. Res., 99, 12767–12771, 1994.

**General comments**
The reasons for some of the choices made in creating the OFAM model configuration are puzzling to me, and need to be explained clearly in the paper. The development of a global ocean-only model whose grid reaches 75°S (most of the Antarctic coastline) and has biology but no ice model, and which excludes most of the Arctic, leads to potentially severe limitations on the general applicability of the model to climate studies, and needs to be clearly justified. Such a model may well be useful for understanding recent climatic variability in the mid latitudes and tropics, providing that enough confidence exists in the forcing dataset used (which is not the case before the satellite era), but cannot inform about the interactions between the high and mid latitudes, and certainly can say nothing about

the climatically highly important polar regions. For this reason I believe it will have limited interest for the climate science community.

> We agree with the reviewer that our mode domain and lack of a sea-ice model prevents this study from being used for interactions between the high and mid latitudes. However, we would argue our study is useful for many other research topics of climate science, as climate science is not necessarily limited to studies on the global scope. There is a need for climate impact studies at regional or national scales. The global warming hiatus over the recent decades, as one example of "hot" climate research topics, has been popularly explained by changes in the tropical and subtropical oceans.  Another example is how the western boundary currents respond to climate change – a question that requires a model realistically represents the boundary currents and eddies there.
>
> The immediate application of the model experiment is to understand the ocean's response to decadal climate variability and climate change. Based on the responses to this manuscript and recent seminar/conference presentations, there is plenty of interests from climate science community in our results. Please also see our General Response #1. A similar near-global model domain ($75^o$S to $75^o$N) is used by the OGCM for the Earth Simulator (OFES) developed by JAMSTEC. The OFES model simulation is still one of the most popular eddy-resolving modelling to do climate-related studies (e.g., Masumoto et al. was cited for 330 times).  We do think it is appropriate to recommend that "the climate modelling community should consider adopting the approach described in this study as **an efficient and short-term solution**" to use high-resolution models ($1/10^o$ and finer) to investigate the climate variability and change on tropical and subtropical oceans.

Would it not have been more logically consistent to develop the model from the start as a coupled system, including a fully global ocean? A non-global ocean model cannot directly be reframed as a free-running coupled climate model.

> In Australia, there is ongoing development of a climate model with $1/4^o$ ocean component.  The development of a coupled climate model with $1/10^o$ ocean component is a long-term goal, subject to substantial increase of computation resources in the future. We started with this near-global OGCM and gained useful experiences in running it for climate applications. We have plan to set up a true global OGCM with sea ice model in coming years.

The Discussion and Summary Section mentions the intention of re-running the model with the merged out-put of an ensemble of CMIP5 coupled models, instead of the JRA55 forcing dataset. This is, of course, perfectly feasible, but would have the severe disadvantage of omitting the interannual and decadal climate variability, which would be averaged out in the ensemble mean.

> The atmospheric forcing for future run is generated by combing high-frequencies (cut-off period of 7 years) from JRA-55 reanalysis product with the long-term climate change from the CMIP5 ensemble. The climate variability forcing from JRA-55 reanalysis has been intentionally included exactly for the purpose of including climate variability as pointed out by the reviewer. We are currently finishing a draft manuscript on the future run, including details about how we prepared atmospheric forcing, but it's a separate paper and out of the scope of this one.
>
> Please also see our General Response #2 and Fig. A for more information.

The method of applying the fluxes equivalent to the initial drift of the model is evidently effective in this case in reducing drift on decadal timescales, and is an interesting way to address the thorny question of spinup in climate models.

> Thank you for your positive and insightful remarks about our method.

This has of course possible drawbacks, however: specifically, an implicit assumption that the drift is due to model deficiencies, rather than to errors in the applied surface fields or in the way the lateral boundary conditions are applied. In particular, the need for strong restoration of temperature and salinity at the northern boundary has serious implications for the performance of an eddy resolving

ocean model, since there is an abrupt mismatch between the resolutions of the model grid and of the forcing data: the consequences for the representation of boundary currents, which are vital for the exchange between the Arctic and the rest of the oceans, are not clear, and should be discussed.

The reviewer is right about the implicit assumption about the model drift and we have now emphasized in the text that this study is not suitable for studying the exchange between the Arctic and the rest of the oceans due to model limitations. We have incorporated the following information into the revised manuscript to explicitly acknowledge the limitations of using an artificial northern boundary in the North Atlantic.

"To ameliorate the impacts of northern boundary, we designed a $5^o$ buffer zone which has linearly varying relaxation time scale, increasing from 30 days at the boundary to 365 days in the interior. The "buffer zone" technique has been commonly adopted in many existing basin or near-global experiments, e.g., Smith et al. (2000) used a $3^o$ buffer zone at $72.6^oN$ in their $1/10^o$ North Atlantic basin model; Masumoto et al. also used a $3^o$ buffer zone in their $1/10^o$ near-global model domain from $75^oS$ to $75^oN$. The buffer zone method comprises the use of the model simulation in the North Atlantic for circulation processes that influenced by the exchanges across this boundary like the North Atlantic Deep Water (NADW) formation."

I have difficulty with the recommendation at the end of the paper, namely that the climate modelling community "**can consider adopting the approach described in this study as an efficient short-term solution, at the same time also develop more sophisticated methods to address this important problem of model drift**". This method was indeed often used in the coupled climate models used at the end of the last century, and was called "flux correction", though this was mainly (but not exclusively) restricted to surface fluxes. By around 2000, though, climate models had improved to a state where they had realistic enough surface fields that they were able to be integrated without flux correction (for example HadCM3, Gordon et al, 2000). Admittedly, interior drifts remain an issue in coupled models, but work is ongoing to reduce this; for instance the use of isopycnal-type ocean models (e.g. Megann et al, J.Clim 2010, Dunne et al J.Clim 2011), that reduce spurious numerical mixing. Do the authors consider that interior drift is a serious enough problem to merit such an invasive engineering fix? This is not dis-cussed in the paper. In my opinion, the proposed use of flux correction in standalone climate models would be a serious step backwards.

We agree with the reviewer that significant progress has been made in coupled climate models to address climate drift. It would be ideal to tackle the interior drift by more scientific techniques other than our "engineering solution". The artificial drift is a still a challenge in the "state-of-the-art" climate models as shown by Sen Gupta (2013) for CMIP5 models. Their study showed drift is still a problem especially in the deep ocean where the drift is comparable to the forced response. Please also see our General Response #2. The drift problem is even more challenging for eddy-rich ($1/10^o$ and finer) simulations where we lack the computational resources to do multi-century simulations.

Clearly, we do not claim our model and approach is a "perfect" solution. On the contrary, our approach should be considered as an efficient and practical approach in the short term. Modelling groups need to develop better models with improved model physics, which do not suffer from climate drifting in long integration (> 50 years).

In any case, the question needs to be posed of the robustness of the tuning of the correction fluxes: would they still be appropriate for models used for future climate projections?

The same climatological restoring (correction) is repeated every year so that it doesn't contribute any forcing to the ocean on interannual and longer time scales. The climate change signals, such as ocean warming associated with anthropogenic climate change, can still be allowed to penetrate to the deep ocean. Such repeated climatological correction has been applied by others over long integration period (e.g., Ding et al. 2015), and our methodology is comparable to their approach. Whether the correction is still appropriate for future climate projections is worth further investigation, but it's beyond the scope of this paper.

For publication in GMD, the intended application of the model described should be clear, and I am not convinced that is the case here: the introduction section of the paper does not make the case strongly enough for the utility of the OFAM model in climate studies.

We modified introduction to clarify the intended application of this model. Please see our General Response # 1 in the beginning. The model is most suited for looking at climate variability in boundary currents and how their changes affect the meso-scale eddies they generate.

In particular, I do not really understand the inclusion of the word "forecasting" in the name of the model, since it is incapable of advancing beyond the limit of current forcing datasets. I would certainly like to see a clearer exposition of what the model is actually useful for.

Obviously our current model experiment is not about "forecasting", but the underlying model is adapted from the Ocean Forecasting Australian Model, version 3 (OFAM3, Oke et al. 2013). The model was primarily developed for short-range operational reanalysis and forecasting. "Forecasting" is used only when we referred to the name of the model (OFAM3) in the Introduction Section. To be consistent with previous literature we would like to retain the name OFAM3 for the model.

**Recommendation**

In summary, although the paper is overall well structured and well written, the lack of clear justification for the design of the model, nor of any clear, plausible statement of its intended application, means I cannot recommend publication in GMD of the manuscript in its current form.

We appreciate your positive remarks, and addressing your comments point by point has helped us improve our manuscript significantly. In particular, we revised the Introduction section significantly to state clearly the design and application of our model, and the Discussion and Summary Section to discuss implications of this work. We hope you will find our revised manuscript is now suitable for publication in GMD.

---

## Author Comment (AC3) · 7 Jun 2016

**Responses to Reviews of manuscript "A near-global eddy-resolving OGCM for climate studies" (gmd-2016-17) by** X. Zhang, P. R. Oke, M. Feng, M. A. Chamberlain, J. A. Church, D. Monselesan, C. Sun, R. J. Matear, A. Schiller and R. Fiedler.
(The reviewer's comments are in back and our responses are in blue)

**Referee Comment by referee #2 (S. M. Griffies)**

This manuscript provides a summary of features found in a mesoscale eddy rich ocean simulation forced by JRA55 atmospheric reanalysis. The manuscript is well written and offers useful diagnostics for others to compare/contrast. It is suitable for GMD, and its publication should ultimately occur. However, there are some overall minor changes needed to bring the manuscript into a more suitable format. So long as the authors address all reviewer comments, and I trust they can, then I recommend this work be published in GMD.

> Thank you for your positive general comment above, and detailed comments below. We are going to address your comments point by point below.

**General comments:**
**Remove "eddy resolving" everywhere
I strongly rebel against the term "eddy resolving". That term is not justified here, nor even defined. The ocean mesoscale, as defined by the 1st baroclinic Rossby radius, has a non-homogeneous eddy length scale spanning from 1km on the shelves of the high latitudes, to 100km in the low latitudes (see Figure 1 in Hallberg, Ocean Modelling 72 (2013) 92–103). So even from the 1st Rossby radius perspective, this model is not "eddy resolving" everywhere. What about higher baroclinic modes?? And what about submesoscale eddies, and then nonlinear gravity wave "eddies", each of whose features reach down into the sub-kilometre range?
Furthermore, there is no study showing numerical convergence in the mesoscale resulting from a model that has a resolution equivalent to the first baroclinic Rossby wave. What in fact do we need to resolve in order to claim we are "resolved"? Is it just a linear baroclinic wave itself? Or the flux convergences? What fluxes? PV, heat, salt, momentum, etc? The question of what defines "eddy resolving" is not closed, so please avoid this sort of terminology. You have a model that richly represents nonlinear mesoscale features, and you are exploring elements of the simulation. But you cannot claim to be "eddy resolving" by any stretch. Period.
So please drop the pretentious and ill-defined "eddy resolving" term ***everywhere*** in your manuscript. Instead, be more explicit and honest by using language such as "mesoscale eddy rich".

> "Eddy-resolving" and "eddy-permitting" are used to "label" the ocean models by some authors. For example, Marsh et al (2009) directly referred to their $1/4^o$ models as "eddy-permitting", and $1/12^o$ models as "eddy-resolving".
>
> We fully agree with your comment above, and in the revised text, we have avoided using the "eddy-resolving" term and use the term "eddy-rich" or "$1/10^o$" instead.
>
> Reference:
> Marsh, R., and Coauthors: Recent changes in the North Atlantic circulation simulated with eddy-permitting and eddy-resolving ocean models, Ocean Modelling, 28, 226–239, 2009.

**Why no sea ice model?? It needs to be better motivated, even if it is the result of a "model of opportunity" exercise.
> We admit that we chose to adapt an available model and designed new strategies to run it. The OFAM3 (Oke et al. 2013), based on MOM, did not include sea ice because it was mainly designed for mid and low latitude studies, which excluded the Arctic and avoided the complication of sea ice. We started with this near-global OGCM and gained useful experiences in running it for climate applications in non-polar regions. We plan to set up a true global OGCM with a sea ice model in coming years.

**change "and also" to "and" throughout the paper. page 2, line 12 page 7, line 17 page 12, line 22
 Corrected.

**Specific comments:**

page 3, line 12: "ever" should be "even"
 Corrected

page 3, line 23: "Saramiento" should be "Sarmiento"
 Corrected

page 4, line 9: The MOM code uses partial bottom cells based on the work of Adcroft et al (1997) as well as the following paper, which should also be cited: @Article{PacGnan1998, author = "Ronald C. Pacanowski and A. Gnanadesikan", title = "Transient response in a $z$-level ocean model that resolves topography with partialcells", journal = "Monthly Weather Review", year = "1998", volume = "126", pages = "3248-3270" }
 Reference is added

page 4, lines 18-20: Boussinesq models need not retain a constant volume. They are quite able to increase or decrease sea level through water addition or removal. For example, see the paper @Article{Lorbacher_etal2012, author = "K. Lorbacher and S. J. Marsland and J. A. Church and S. M. Grifn/fn/ies and D. Stammer", title = "Rapid barotropic sea-level rise from ice-sheet melting scenarios", journal = JGR, year = "2012", volume = "117, C06003", doi = "doi:10.1029/2011JC007733", }
The reason that groups often keep the net water flux equal to zero over the globe is to reduce model drift. It has NOTHING to do with the Boussinesq approximation.
 You are right about Boussinesq models need not retain a constant volume. Our writing was a bit confusing. The sentence is now modified to
 "The model adopts Boussinesq Approximation (Greatbatch 1994). The net global freshwater flux (i.e., sum of evaporation, precipitation and river run-offs) is balanced at each model time step, thus the global mean sea level is kept constant."

 We chose to balance the freshwater flux at each time step mainly because the quality of freshwater flux is insufficient to ensure a realistic temporal evolution of the global mean sea level. Moreover, reliable fresh water fluxes from melting of glaciers and ice sheets are not available and therefore not included in most OGCM experiments. Since we are mainly interested in dynamic sea level (i.e., deviation of regional sea level from the global mean), we chose to keep the global mean sea level constant by having no net freshwater flux. The zero net freshwater flux to the ocean was not set up for controlling model drift (in the deep ocean).

page 6, line 8: What is "The correct spin-up of OGCMs"? Do you presume to have example of such? I for one have been on a 25 year quest for this method... :-)
 As both you and referee #1 pointed out, it's really a challenging question. It may require tailored treatments for different model experiments. Here, we intended to strengthen the importance of "correct spin-up". The methodology we used in this paper may be one of feasible solutions to a similar set-up. The sentence and terminology are modified now.

page 7, line 13: I prefer to think of the global OHC as the global volume mean heat content. However, as used here, the authors refer to the horizontally integrated heat content as a function of depth. I suggest a more suitable language is useful to avoid reader confusion.
 It's changed to "ocean heat content – integrated both vertically over some depth ranges and spatially over all ocean areas".

page 7, line 14: "staring" should be "starting"

page 7, line 18: there seems to be a missing number on this line. It presently reads "change of Global OHC...". The blank must be a number, but that number is missing.

It's not due to missing number. It's modified to "the global OHC change rate".

Figures: I encourage placing more statistical information on each of the figures or their captions, so to better allow them to be self-contained. Many statistics are noted in the text, and they should also be placed along with the figure. Additionally, for the maps, it would serve the reader well to also provide a zonal mean of the biases to better identify the latitude where the biases are localized. All of this information is useful for others aiming to perform quantitative comparisons to your work. Merely showing the maps is insufficient.

Statistical information is added to figures now. Zonal mean of biases is also added to Figs. 8 and 9.

page 9, line 19: "indicates" should be "indicate"

Corrected

page 9, line 24: JRA-55 is available at a resolution of 55km as well as 1.25 degrees. http://rda.ucar.edu/datasets/ds628.0/. You can examine this hypothesis concerning upwelling by moving to the finer resolution data.

That's a good suggestion. It could be a nice future work.

page 10, section 4.1.4: Please show the maths for how you computed the Sverdrup/Island rule transport shown in Figure 10. Others may wish to repeat this calculation, and your method should be clearly documented. Reliance on literature is not needed, since you can summarize the method in a few lines of words and maths.

Details about the Sverdrup Balance and Island rule calculation are given in the Appendix B now.

page 11, line 2: The meridional overturning streamfunction is not "zonally averaged", in which case the units would be m2/sec. Instead, it is "zonally integrated". Please correct.

Corrected

page 14, line 13: More discussion is needed regarding this point. It contrasts with points that others have made. Namely, even though OISST is 1/4 degree, it is sampling the real ocean with real ocean variability. This situation is quite distinct from a 1/4 ocean model that only admits variability at the 1/4 and coarser range. For more on this point, please see the paper Levy et al, Ocean Modelling, 48, 2012, Pages 1–9.

Point acknowledged.
Text is modified to state that the model is still missing unresolved sub-mesoscale processes that could reduce the variability simulated.

page 15, line 9-10: I am unsure why the bulk formula should produce less wind stress than the wind stress directly provided by JRA55. What are the issues? Perhaps it is due to use of (U_atmos - U_ocean) to compute the stress??

The model uses (U_atmos - U_ocean) for the calculation, which plays a role in wind stress difference. Additionally, although both JRA55 and our model use bulk formula based on the Monin-Obukhov similarity theory, the implementations are not identical.

page 15, line 30: "Domingues"

Corrected

page 16, line 3-4: I have trouble with your statement that "no observational data are assimilated in our historical experiment." And later the statement "nearly free OGCM". Your results are an achievement. But there are huge limitations.
–What about the non-adaptive forcing in the ocean interior that reduces drift (page 16,

line 20)?

–What about the adaptive surface nudging?

–What about the truncated northern and southern domain boundaries?

–What about the absence of sea ice?

All of these limitations are helpful to reduce model drift. The high latitudes in particular are very very tough in ocean-ice models. See the Griffies et al, 2009 CORE paper for mechanisms leading to drift, with these mechanisms largely eliminated by truncating your model domain and removing sea ice.

> The sentence is revised to
>
> "This is a good indication that the model represents change and variability of OHC realistically, although it's not an ocean reanalysis product like SODA (Carton and Giese, 2008) and ECMWF ORAS4 (Balmaseda et al. 2013), both of which assimilate massive in-situ and satellite data. The surface flux adjustment and non-adaptive restoring in the deep layers, as well as nudging in the northern boundary, prove to help achieve the good comparisons."

page 16, line 29: "...associated with the..."

> Corrected

page 18, line 11: Suggested edit: "Despite good model performance for certain diagnostics/metrics, we fully acknowledge many caveats and limitations."

> Corrected

page 18, line 17: "combing" should be "combining"

> Corrected

page 27, line 4: "annual"

> Corrected

page 28: this table needs observational estimates for the reader to have any idea of how relevant the model transports are!

> Observational estimates are added.

page 29, Figure 1: I can barely read the numbers places on the map. They will undoubtedly come out even less visible in print.

> The readability of Fig. 1 is improved now.

page 31, Figure 3 caption: What does "Equivalent surface heat flux" referred to here? Please define.

> Revised to "ocean heat content change rate"

page 32, Figure 4: I suggest rotating these panels clockwise by 45degrees, so that a full profile for temperature and salinity sit atop one another.

> Revised as suggested

page 33, Figure 5: Again, what is "equivalent surface heat flux"???

> Revised to "ocean heat content change rate"

page 34, Figure 6 caption: penultimate domain should be "north subtropical", not "south".

> Corrected

page 35, Figure 7: place zonal mean bias next to bottom panel, and make note of other statisticis in the figure caption.

> Revised as suggested

page 36, Figure 8: difference map should have smaller colour range than raw fields, in order to better view the biases. Also add statistics information to the caption.

> Revised as suggested

page 41-42, Figures 13,14: The BGC figures use black land mask, whereas other figures use grey. Please be consistent.

Revised as suggested

page 50, Figure 22 caption: "Domingue"

Corrected

---

## Author Comment (AC4) · 7 Jun 2016

**Responses to Reviews of manuscript "A near-global eddy-resolving OGCM for climate studies" (gmd-2016-17) by** X. Zhang, P. R. Oke, M. Feng, M. A. Chamberlain, J. A. Church, D. Monselesan, C. Sun, R. J. Matear, A. Schiller and R. Fiedler.

(The reviewer's comments are in back and our responses are in blue)

**Referee Comment by anonymous referee #3:**

I find little merit in this manuscript and feel it is unsuitable for publication. The paper describes a collection of ad hoc fixes and patches to the forcing of a high-resolution global model to compensate for shortcomings in the formulation of the problem (nonglobal, lack of sea-ice). The methods described are typical of those used in the 1980's and 1990's when available surface forcing products were far inferior. The apparent goal of the authors is to share "Experiences gained from our numerical experiments : : :" (pg 2 line 15). I cannot agree that this experience is something any other group should be following in the 21st century.

> Please see our General Response #2 in the beginning about using "old" technique, hope you are convinced that there is still some room to use them wisely.

> Also we want to point out that the OGCM for the Earth Simulator (OFES, Masumoto et al. 2014) developed by JAMSTEC used the same near-global domain (75S to 75N) as ours. The OFES model simulation is still one of the most popular eddy-resolving modelling to do climate-related studies Their model is neither fully global nor includes sea ice, but it is considered useful by the climate community.

There is no innovation in ocean model development described in this paper. There are some changes made relative to the model formulation described in the previous paper by Oke et al (2013), but none of these are new to the modeling community (e.g. the incorporation of the KPP mixing scheme). Further, the authors do not described the sensitivity of the solutions to these changes in formulation.

> We want to clarify that this manuscript was submitted as a "Model experiment description paper", rather than a "Model description paper" or "Development and technical paper", according to the GMD guideline at http://www.geoscientific-model-development.net/about/manuscript_types.html. This paper is not about innovation in ocean model development, but about describing a model experiment based on an existing model. In particular, we developed new strategies to run the model so that it can be used for climate studies.

> We modified the sentence to explain our usage of KPP mixing scheme.

The main argument for "innovation" in the manuscript is in the adjustment of the surface forcing to constrain global energy balance and addition of fictional forcing in the deep ocean to limit drift. The procedures are described as they were implemented without discussion of their merits relative to other alternatives, nor of why the particular choices were made. In the case of the surface flux adjustment, an obvious and simpler method would be to assess the global imbalance in the JRA-55 forcing using observed SST in the bulk formula prior to model integration and subtract the necessary correction a priori. What is the advantage of the iterative approach described in the paper? Why was the adjustment applied to long wave down flux? Why not reduce shortwave down? There is no rationale provided for the choices made.

> Please see our General Response #1 in the beginning about our motivation of this model experiment. For both constant flux adjustment and non-adaptive restoring in the deep ocean, we have given enough details about their purposes and procedures.

> For surface heat flux adjustment, we could adjust either downward longwave radiation or downward shortwave radiation. We chose to adjust longwave for its easy implementation. The results would not be sensitive if we chose to apply the small adjustment to the downward shortwave radiation.

> For the iterative approach, please see our response below related to your comment about pg6 /line 27-bottom.

In the case of the deep restoring, the authors imply that this is common practice in high resolution modeling, but only reference their own work as an example. I cannot think of a single study since the pioneering work of Semtner and Chervin in the 1980s that has used deep restoring in a forward integration of a high-resolution model. The main "innovation" the authors claim is the introduction of "non-adaptive" restoring. This is not a new idea. The same basic ideas are described in for example Eden et al JPO, 34, 701-719, 2004 (and likely a number of other papers).

> About non-adaptive restoring in the deep ocean, we give some detailed explanation in General Response #2 in the beginning, and also gave one example (Zhang et al. 1998) below related to your comment about pg4 /line 24. Eden et al. (2004) is also cited now.

Beyond the generally poorly motivated paper, the manuscript is quite superficial in its assessment of the solutions, and largely speculative when it comes time to attribute bias. Phrases like "may explain" or "could indicate" appear throughout when a more quantitative evaluation at a process level is called for. The aspects of the manuscript dealing with the BGC simulation seem to have been added as a complete afterthought.

> We add more quantitative discussion now, and further discuss the motivation for including BGC in this experiment.
>
> We are particularly interested in how the biophysical coupling is affected by mesoscale dynamics. As mentioned in the introduction, previous results indicate that mesoscale processes can modify the mean state. By including initial BGC results here we are preparing for further analysis, not just of the historical period, but also future climate projections.

**DETAILED COMMENTS**

pg 4 / lines 14-16. What is the implied global net heat flux for the JRA-55 product when using observed SST? This is a direct indication of the expected drift.

> The usage of bulk formal to calculate heat flux on the fly makes the heat flux applied to the model ocean depends on ocean solution, including its global net heat flux. For example, in the Coordinated Ocean-ice Reference Experiments (COREs), a common atmospheric forcing dataset is applied to 7 global ocean-ice models, the global net heat flux is different among them (Griffies et al. 2009). In other words, the flux correction would be different for different models, even driven by the same atmospheric dataset. Therefore, in our experiment, we used this pragmatic method to derive the required flux correction, which was determined by the interaction between our model and JRA-55 forcing.

pg 4 / line 19: Why should the hydrologic cycle be balanced instantaneously? Cannot water be stored on land seasonally?

> The balance of hydrologic cycle is not adopted because of model formalization. Our model can handle unbalanced freshwater flux.
>
> We chose to balance freshwater flux at each time step, mainly because the quality of freshwater flux is insufficient to ensure a realistic temporal evolution of the global mean sea level. Since we are mainly interested in dynamic sea level (i.e., deviation of regional sea level from the global mean) rather than the global mean sea level itself, we chose to keep the global mean sea level as zero by balancing freshwater flux.

pg 4 / line 19: Global volume conservation is unrelated to the Boussinesq approximation

> The sentence is modified to "The model adopts Boussinesq Approximation (Greatbatch 1994). The net global freshwater flux (i.e., sum of evaporation, precipitation and river run-offs) is balanced at each model time step, thus the global mean sea level is kept constant."

pg 4 / line 24: if relaxation is "often used" provide some references of examples other than your own model

One example is Zhang J. et al. (1998), who specifically compared modelling results with and without climate restoring in their ocean-ice modelling experiment in the Arctic. They found that restoring of temperature and salinity has significant impact on prediction of the ice–ocean circulation in the Arctic. In particular, restoring salinity and temperature in the deeper ocean can reduce climate drift in the Arctic. This sentence is deleted because of restructure of Sections 1 and 2.

Reference:
Zhang, J., Hibler, W. D., Steele, M., and Rothrock, D. A.: Arctic ice–ocean modeling with and without climate restoring, J. Phys. Oceanogr., 28, 191–217, 1998.

pg 4 / line 33 : it is not a common technique (as above - provide some references)

See above

pg 5 / lines 8-22: This is a slew of no-conservative physics. There needs to be a fuller demonstration of its impact not just on global measures, but on the local structure of the solution. What is the spatial distribution of the restoring term? What spurious heat transports are implied by it? How does it impact the mesoscale?

Based on raised questions, I think you meant lines 8-12 (not 8-22) about restoring in the buffer zone near the northern boundary in the North Atlantic.

To minimize negative impacts of northern boundary, we designed a $5^o$ buffer zone which has linearly varying relaxation time scale, increasing from 30 days at the boundary to 365 days in the interior. "buffer zone" technique has been commonly adopted in many existing basin or near-global experiments, e.g., Smith et al. (2000) used a $3^o$ buffer zone at $72.6^oN$ in their $1/10^o$ North Atlantic basin model; Masumoto et al. also used a $3^o$ buffer zone in their $1/10^o$ near-global model domain from $75^oS$ to $75^oN$.  So our set-up is in alignment with many existing studies.

Although it's natural to think that global model should do a better job than regional model (with boundaries), it's not always the case. For example, Maltrud and McClean (2005) found that the performance of their $1/10^o$ global model is worse than a North Atlantic regional model (same model grid as the global model, but truncated to $20^oS$ to $72^oN$) in representing the Gulf Stream/North Atlantic Current System. Although they didn't give clear conclusion about the contrast between global and regional models, but they suggested that better water mass conditioning through restoring at boundaries ($20^oS$ and $72^oN$) could play some role.

pg 5 / line 8 : what is the "correct" spin-up. The ocean state in 1979 was not in equilibrium with the forcing in 1979.

I think you meant pg 6 (not 5).

How to spin up a model is a challenging question. It may require tailored treatments for different model experiments. Here, we intended to strengthen the importance of "correct spin-up". The methodology we used in this paper may be one  of feasible solutions to a similar set-up.  The sentence and terminology are modified now.

The idea behind normal-year forcing experiment as proposed in CORE-I experiment (Griffies et al. 2009) is to run the ocean model with repeated normal year forcing (1979 in our experiment) long enough so that the ocean can finally reach equilibrium with forcing. Our model experiment basically follows the design principle of CORE-I.

pg 6 / line 15-20: How is the year-end discontinuity handled?

We didn't do special treatment on this. Following the protocol of normal-year forcing experiment by CORE-I (Griffies et al. 2009), we simply repeated the forcing year after year. We didn't notice any obvious discontinuity in our model fields (e.g., see Figs. 2 & 3).

pg 6 / lines 27-bottom: There is no justification provided for this ad hoc procedure. (see above)
This procedure is ad-hoc, but driven by the intention to make the heat imbalance smaller and stabilized as quickly as possible. We are interested in examining heat uptake and distribution in recent decades simulated by this model. The observation indicates a net heat imbalance on the order of ~0.5 W/m2. If we didn't adjust, the ocean absorbs heat at the rate of ~5 W/m2 (ten times of realistic magnitude) over at least the first several years, which leads to a much warm-biased ocean for us to start our historical experiment.

So the main motivation was to reduce excessive heat uptake during the spin-up, though the way how it's implemented isn't so critical. For example, alternatively, we could adjust during Year 3 to 5, then maintain the correction from Year 6 (rather than adjust during Year 3 to 7 as done in our experiment)

pg 7 / line 12 "does not necessarily imply a net heat flux correction" Of course it does. That is what you have constructed it to do!
If the model is forced with flux form forcing, any heat flux correction will be directly passed to the ocean. But if the model is forced with bulk formula (heat flux is calculated on the fly and thus depends on model state, rather than being provided), as implemented in our model experiment, some air-sea feedback can take place, which will affect heat flux. About this point, refer to Table 1, since Year 7, the model is repeatedly driven by the same atmospheric forcing through bulk formula until Year 20, but the annual net heat flux keeps changing (gradually decreases).

pg 7 / line 32-35: Why does it agree better later in the run? The SST has diverged further from the observed initial conditions. You should be comparing to OAflux for 1979, not its climatology.
The initialized temperature field (including SST) is from the CARS climatology (Ridgway and Dunn 2003) based on data collected over the recent decades, not from the 1979 ocean state. So the SST evolves from its initial climatological value, partially because the model is adjusting to reach equilibrium with the forcing in 1979. In the first several years, the ocean is still in the initial shock of spinup and a large and unrealistic heat imbalance is not a surprise.

OAFlux product (1983 to current) doesn't have data in 1979, thus climatology is plotted instead.

pg 9 / line 12 : "model may be too efficient" This is not a meaningful diagnosis and is purely speculative. Could be equally well attributed to any other process.
Determining the underlying processes needs some further investigation, such as the mixed layer heat budget. For this paper, we are trying to suggest (rather than diagnose) possible reasons by considering ocean dynamics and existing studies.

pg 9 / line 18: "This systematic difference : : :" The biases in the Gulf Stream and Kuroshio appear well to the north of separation points, an indication of poor boundary current separation. This is consistent with the biases int he SSH as well.
Thank you for your suggestion. The poleward position of the separation point in the model, relative to observations, is a factor. We now add some discussion on this.

pg 9 / line 29: "Their global means are almost the same : : :" By construction – you formulated to model to keep global mean sea level constant!
This sentence is deleted now.

pg 10 / line 28 "we do not repeat the detailed comparison" Then you don't need the detailed Table.
The sentence is revised. We also added observation-based transports for comparison, upon suggestion by Referee #2.

pg 11 / line 10: A more useful and critical comparison would be against the vertical structure of the RAPID overturning.

As the reviewer suggested, we compare the overturning stream function at 26°N in the Atlantic between the model and RAPID Array observation (McCarthy et al. 2015), over the overlapping period 2004-2014 (Fig. B below). The maximum of the stream function, often referred to the AMOC, is 15.9+-2.4 SV from the model, close to 16.9+-3.5 from the RAPID Array. The depth of maximum stream function is around 1000 m in both model and observation. The temporal correlation of AMOC between model and observation is 0.66 over 2004-2014, but jumps to 0.81 over the later period (2009-2014). Fig. B is now included in the manuscript.

[Figure]

**Figure B.** Overturning stream function (Sv) at 26°N in the Atlantic from (a) the model and (b) the RAPID Array observations over 2004-2014.

Reference:
McCarthy, G., and Coauthors, 2015: Measuring the Atlantic meridional overturning circulation at 26°N. Prog. Oceanogr., 130, 91–111, doi:10.1016/j.pocean.2014.10.006.

pg 11 / line 27 : They appear more dissimilar than similar to me. The model is completely missing most of the upwelling productivity and vastly overestimates the equatorial productivity.

We acknowledge similarities of the simulated BGC fields to observations are not as good as similarities of physical metrics (such as SST and SSH). Correlations of simulated BGC fields with observations are challenging due the fields being sensitive to both physical and biological processes and their uncertainties in the simulation, and also due to limitations and uncertainties in the observations.

As a way of comparison, we have calculated correlations of the mean fields shown to the available observations, and compared these to the range of correlations from a recent ocean BGC model intercomparison paper (Kwiatkowski et al 2014). Our correlation of primary productivity is 0.31, compared to a reported range of -0.08 to 0.64, and for carbon flux the correlation is 0.69, compared to a range of 0.01 to 0.68. These are now reported, in the manuscript.

Reference:

Kwiatkowski, L., Yool, A., Allen, J. I., Anderson, T. R., Barciela, R., Buitenhuis, E. T., Butenschön, M., Enright, C., Halloran, P. R., Le Quéré, C., de Mora, L., Racault, M.-F., Sinha, B., Totterdell, I. J., and Cox, P. M.: iMarNet: an ocean biogeochemistry model intercomparison project within a common physical ocean modelling framework, Biogeosciences, 11, 7291-7304, doi:10.5194/bg-11-7291-2014, 2014.

pg 13 / line 21 : as above - they are more dissimilar than similar. The largest observed variability is in the upwelling areas, not the central Pacific.

The position of BGC variability is related to gradients in the mean fields, which have biases that are larger than physical biases, but are reasonable relative to other BGC simulations, as discussed above.

10

pg 14 / line 14: "its possible that : : :" There is no need for speculation here. Smooth the model solution to the length scales used in the OI procedure for the observations and compare the smoothed fields.

15

Smoothing model SST (from $1/10^o$ to $1/4^o$) doesn't change the result too much, still higher than the observation. In response to suggestion by Referee #2, we have added a comment to indicate that the model variability may be related to forcing, or due to unresolved, sub-mesoscale processes which are found to lower variability (Lévy et al. 2012).

20

pg 14 / line 22 : How can the mean upwelling be simulated poorly, but the variability be well reproduced?

Representation of mean and variability in models isn't necessarily tied with each other. Good representation of mean doesn't imply good representation of variability, and vice versa (e.g., Bates et al. 2012). Later in the paragraph, we did point some locations where model is different from observation. This sentence is revised.

25

Reference:
Bates, S. C., B. Fox-Kemper, S. R. Jayne, W. G. Large, S. Stevenson, and S. G. Yeager,: Mean biases, variability, and trends in air–sea fluxes and sea surface temperature in the CCSM4. J. Climate, 25, 7781–7801, 2012.

30

pg 15 / line 15: What are we supposed to be seeing in the single-month plots? The eddies will all be in different places for a forward model run.

35

The reviewer is right. Eddies from model and observation at any time aren't identical due the "chaotic" nature of eddies. We can only compare them by statistical measures, like long-term mean in Fig. 21a, b. Here we give the monthly snapshots to show the model can simulate a similar eddy distribution in those eddy-active regions (like western boundary currents) as observed by satellite altimeter.

40

pg 15 / line 31 - : The full depth OHC is completely determined by the surface heat flux, so it is not surprise these are similar. Why are there no comparisons with the vertical structure from the reanalysis products?

The reviewer is right - change of full-depth OHC is determined by the surface heat flux (because the model or the real ocean conserves energy). But the surface heat flux forcing is not necessarily the same for all ocean reanalysis products and our model experiment. In particular, heat flux is calculated on the fly through bulk formula in our experiment, rather than prescribed from atmospheric reanalysis, so the heat flux forcing depends on the model state, and we have no direct control over it.

50

Because the manuscript is already quite lengthy with 26 figures, we are planning a separate scientific paper describing the heat uptake and redistribution, and underlying physical processes based on this historical modelling experiment.

55

Only EKE is derived indirectly from altimetry sea level data. We repeated model EKE calculation using sea level rather than surface velocity, and got very similar results in most areas, except some regions (like western boundary currents) where ocean EKE can't be fully derived from sea level.

---

## Author Comment (AC5) · 7 Jun 2016

**Responses to Reviews of manuscript "A near-global eddy-resolving OGCM for climate studies" (gmd-2016-17) by** X. Zhang, P. R. Oke, M. Feng, M. A. Chamberlain, J. A. Church, D. Monselesan, C. Sun, R. J. Mattear, A. Schiller and R. Fiedler.
(The reviewer's comments are in back and our responses are in blue)

**Short comment #1 by Y. Yu**

The manuscript by X. Zhang et al. presented very comprehensive diagnosis from a high resolution OGCM forced by JRA55 reanalysis, which provided very helpful information for many researchers on modeling, climate change etc. Therefore, it is suitable for GMD journal, and I would like to recommend it to publish.

Thank you for your nice comment above, and detailed comments below. We are going to address your comments one by one below.

Here are some detailed comments as follows.

1. The mechanism of 1998-2004 Hiatus remains unclear, the numerical experiments in this manuscript provide an important opportunity to understand the Hiatus. For example, how does the model reproduce the basic characteristics of Hiatus such as temperature anomalies in the surface, subsurface and deep ocean, and is there any relationship between hiatus and AMOC?

It is a very good suggestion. In fact, we are planning a separate scientific paper describing the heat uptake and redistribution, and underlying physical processes based on this historical modelling experiment, in particular related to heat hiatus period.

2. As you mentioned in page 4, bulk formula as suggested by Large and Yeager (2004) are applied to calculate the turbulent heat flux and moment flux at the sea surface? However, I am very curious that you have to apply for a large heat correction more 16W/(m*m). Is it resulted from overestimated downward radiation flux in JRA55?

We chose to modify downward longwave radiation for the ease of implementation. Since bulk formula is used to calculate heat flux, and some air-sea feedback mechanisms can lead to compensation among different components. In fact, in our model experiment, reduced downward long wave radiation is mainly compensated by less heat loss through latent heat flux (see Fig. C below).

3. Because of large heat flux correction, I suggested that the authors had better show comparison of zonal mean shortwave and longwave radiation from JRA-55, OAFLUX or other available observation.

Upon your suggestion, we compared the zonal mean heat flux component from our model experiment with ECMWF Interim Reanalysis and OAFlux, and found the comparison is quite good (See Fig. C below). Nonetheless, the comparison is better with Interim Reanalysis, since OAFlux has a (unrealistically) large positive mean global net flux around 30 $W/m^2$. The upper panel of Fig. C is now included in the manuscript.

[Figure]

[Figure]

**Figure C.** Zonal mean heat flux (W/m$^2$) from our OFAM3-JRA55 model run and ECMWF Interim Reanalysis (upper panel) and OAFlux (lower panel). All flux components and net heat flux are defined positive when the ocean gains heat.

4. How is the temperature change defined in the Figure 4? Is it defined as difference in temperature between the last day and the first day for a given model year?

It's defined as the difference of temperature in the first day between two adjacent years. For example, change over Year 1 is defined as value on the first day of Year 2 minus that on the first day of Year 1, and so on. Information is added.

5. Figure 7, there are significant warm bias at high latitudes in the North Atlantic. Is it due to surface boundary condition, lateral restoring boundary condition or something else?

This warm bias needs further investigation, since it may be caused by several factors. The warm bias off US & Canada east coast may be associated with more poleward location of the Gulf Stream. The warm bias in the Labrador Sea isn't necessarily associated with the northern boundary in our model, since Marzocchi et al. (2015) found similar warm bias in this region in a 1/12$^o$ global OGCM.

6. The caption in Figure 9 is "Mean Eddy Kinetic Energy ...", is this correct?

It's a typo, which should be "Mean Kinetic Energy". Corrected.

7. Figure 10, I suggested that "mean stream function" in the caption should be replaced with "mean barotropic stream function".

We prefer not to modify, since this figure also contains results from the Sverdrup Balance & Island rule, which, strictly speaking, are not barotropic solutions.

8. The authors calculated simulated MKE and EKE using surface currents, but observed MKE and EKE using sea surface height. If both simulated and observed EKE and MKE are estimated from sea surface height, the comparison may be more reasonable.

Observed MKE is based on surface currents from drifter buoys (Lumpkin and Johnson 2013). Only EKE is derived indirectly from altimetry sea level data. We repeated model EKE calculation using sea level rather than surface velocity, and got very similar results in most areas, except some regions (like western boundary currents) where ocean EKE can't be fully derived from sea level.

5